# Involvement of the p38 MAPK-NLRC4-Caspase-1 Pathway in Ionizing Radiation-Enhanced Macrophage IL-1β Production

**DOI:** 10.3390/ijms232213757

**Published:** 2022-11-09

**Authors:** Ji Sue Baik, You Na Seo, Young-Choon Lee, Joo Mi Yi, Man Hee Rhee, Moon-Taek Park, Sung Dae Kim

**Affiliations:** 1Research Center, Dongnam Institute of Radiological & Medical Sciences, Busan 46033, Korea; 2Department of Medicinal Biotechnology, College of Health Sciences, Dong-A University, Busan 49315, Korea; 3Department of Microbiology and Immunology, College of Medicine, Inge University, Busan 47392, Korea; 4Department of Veterinary Medicine, College of Veterinary Medicine, Kyoung Pook National University, Daegu 41566, Korea

**Keywords:** ionizing radiation, IL-1β, NLRC4, caspase-1, p38 MAPK, macrophage

## Abstract

Macrophages are abundant immune cells in the tumor microenvironment and are crucial in regulating tumor malignancy. We previously reported that ionizing radiation (IR) increases the production of interleukin (IL)-1β in lipopolysaccharide (LPS)-treated macrophages, contributing to the malignancy of colorectal cancer cells; however, the mechanism remained unclear. Here, we show that IR increases the activity of cysteine-aspartate-specific protease 1 (caspase-1), which is regulated by the inflammasome, and cleaves premature IL-1β to mature IL-1β in RAW264.7 macrophages. Irradiated RAW264.7 cells showed increased expression of NLRC4 inflammasome, which controls the activity of caspase-1 and IL-1β production. Silencing of NLRC4 using RNA interference inhibited the IR-induced increase in IL-1β production. Activation of the inflammasome can be regulated by mitogen-activated protein kinase (MAPK)s in macrophages. In RAW264.7 cells, IR increased the phosphorylation of p38 MAPK but not extracellular signal-regulated kinase and c-Jun N-terminal kinase. Moreover, a selective inhibitor of p38 MAPK inhibited LPS-induced IL-1β production and NLRC4 inflammasome expression in irradiated RAW264.7 macrophages. Our results indicate that IR-induced activation of the p38 MAPK-NLRC4-caspase-1 activation pathway in macrophages increases IL-1β production in response to LPS.

## 1. Introduction

Despite the increasing therapeutic success rate of radiation therapy, the resistance of tumor cells to radiation remains a significant challenge and unmet research need. Changes in the tumor environment due to irradiation have been proposed as a potential cause of this resistance [1]. Recent studies underscore the fact that IR can also modulate immune cell functions [2,3,4]. Macrophages are an essential component of the immune cell population in the tumor microenvironment. IR has been reported to stimulate the priming or activation of macrophages in vitro [5]. Irradiation of J774A.1 macrophages at a dose of 20 Gy increased cell activation, accompanied by morphological and enzymatic changes [6]. Irradiation of RAW264.7 macrophages at 50 Gy increased expression of major histocompatibility complex class I antigens and induced greater susceptibility to lipopolysaccharide (LPS), while cytotoxicity was irreversibly primed via tumor cell apoptosis [7]. Collectively, these studies suggest that the immune cell activation induced by IR can influence the outcome of cancer therapy.

Interleukin (IL)-1 is an inflammatory cytokine involved in numerous immune responses such as innate or acquired immunity and plays an important role in controlling infection or aseptic damage [8]. IL-1 is produced as an inactive 31-kDa precursor, termed pro-IL-1β, in response to molecular motifs carried by pathogens. These pathogen-associated molecular patterns act through pattern recognition receptors on the surface of macrophages to regulate pathways that control gene expression [9]. The IL-1 family includes four main members, namely IL-1α, IL-1β, IL-33, and IL-1 receptor antagonist (IL-1RA) [10]. Production of IL-1α and IL-1β was reported to increase upon 2-Gy irradiation to human alveolar macrophages [11]. We also previously reported that IR enhances IL-1β production in response to LPS stimulation in macrophages [12,13]. However, the molecular mechanisms underlying IR-induced macrophage activation remain elusive.

Inflammasomes comprise a family of cytosolic multi-protein complexes that modulate the activation of caspase-1 and promote the maturation and secretion of interleukin (IL)-1β and IL-18 [14]. Precisely, activation of the inflammasome by a secondary signal cuts off IL-1β from the pro-form and induces its switch to the mature form [15,16]. Pro-IL-1β is retained inside the cell, which is subject to proteolysis by caspase-1 to become mature IL-1β secreted out of the cell. Members of the NOD-like receptor (NLR) family can assemble inflammasome complexes with the adaptor protein ASC and caspase-1, resulting in caspase-1 activation and the release of IL-1β. At least three NLR proteins are known to assemble IL-1β-activating inflammasomes directly: NOD-, LRR-, and pyrin domain-containing protein 3 (NLRP3); NLR family CARD domain-containing protein 4 (NLRC4), and NLR family pyrin domain containing 1 (NLRP1). Upon activation, these NLR sensors oligomerize to enable the proteolytic activation of caspase-1 through homotypic CARD–CARD interactions [17,18]. IR exposure can induce inflammasome pathway activation in immune cells [19]. IR activates the NLRP3 inflammasome and induces pyroptosis in bone marrow-derived macrophages [20]. Despite impressive gains, the primary research focus on NLRP3 has left gaps in understanding inflammasome biology. Notably, the mechanism by which inflammasomes become activated remains unclear, and the connections between inflammasome structure and function have not been elucidated to date. However, related studies on another inflammasome inducer, NLRC4, can help to fill these gaps. NLRC4 is a vital component of the inflammasome-mediated response to various microbial stimuli and endogenous danger signals via caspase-1 activation [21]. Mariathasan et al. [22] demonstrated that activation of NLRC4 was likely a ligand-regulated process, and murine macrophages lacking NLRC4 failed to activate caspase-1 after exposure to Salmonella typhimurium. Nevertheless, information on IR-mediated NLRC4 inflammasome activation in macrophages is scarce.

The p38 mitogen-activated protein kinase (MAPK) pathway was originally described as a mammalian homolog to a yeast osmolarity-sensing pathway [23]. p38 MAPK can be activated by multiple exogenous stimuli such as ultraviolet radiation, cytotoxic chemicals, and IR [24]. Activation of p38 regulates a variety of cellular processes such as inflammation, cell cycle arrest, and apoptosis in a cell type-specific manner. For example, p38 MAPK signaling has been shown to both promote cell death as well as to enhance cell growth and survival [25,26,27]. The ability of IR to regulate p38 MAPK activity appears to be highly variable, with different research groups reporting strong activation [28,29], weak activation [30], or no activation [31]. However, there are few reports on the IR-induced activation of p38 MAPK in macrophages [32].

Thus, the aim of this study was to identify the molecular target driving the increased IL-1β production in IR- and LPS-treated RAW264.7 macrophages. Based on the background above, we focused on the potential roles of the NLRC4 inflammasome and MAPK signaling on cytokine production (IL-1β and nitric oxide [NO]) in IR-exposed and LPS-stimulated cells. In addition to the production of the premature form of IL-1β in cells, IR has been shown to activate the immune cell inflammasome and caspase-1 in mice [19], indicating a role in the cleaving of premature IL-1β and conversion to its active/mature form for secretion. We therefore also investigated the effect of IR on the activity of caspase-1, which converts premature IL-1β into mature IL-1β, in RAW264.7 cells.

## 2. Results

### 2.1. IR Enhances NO and IL-1β Production in LPS-Stimulated RAW264.7 Macrophages

We first investigated the effect of IR on the pro-inflammatory cytokine production in LPS-treated RAW264.7 macrophage cells. The experimental schedule is schematically shown in Figure 1. First, the RAW264.7 cells were exposed to 5 Gy of IR and then stimulated with LPS (0.1 μg/mL) 24 h later. After incubation for another 24 h, the supernatant was recovered to evaluate the cytokine levels secreted from RAW264.7 cells by enzyme-linked immunosorbent assay (ELISA).

As shown in Figure 2A,C, in LPS-only-treated RAW264.7 cells, the production of pro-inflammatory cytokines such as NO and IL-1β was increased compared with that in control cells, indicating that the macrophage cell line functions as expected. In addition, no measurable difference in IL-1β production was observed in the irradiated RAW264.7 cells compared to the sham control RAW264.7 cells. Furthermore, IR and LPS-treated RAW264.7 cells showed significantly increased NO and IL-1β production compared with LPS-only treated cells by approximately 690 ± 29% and 1846 ± 98%, respectively. Thus, IR increased IL-1β production to a greater extent than NO production in LPS-stimulated RAW264.7 macrophages. A similar effect was found for *Il1b* expression at the mRNA level based on reverse transcription-quantitative polymerase chain reaction (RT-qPCR) (Figure 2A). Confocal microscopy further confirmed that the intracellular IL-1β level of IR-exposed macrophages was higher than that of control macrophages in response to LPS stimulation (Figure 2D). To confirm that the increase in IL-1β production of macrophages by IR is not a RAW264.7 cell-specific phenomenon, the increase in IL-1β production by IR was reconfirmed in another macrophage cell line, J774A.1 cell (Appendix A) and primary cultured peritoneal macrophage cells (Figure 2F).

### 2.2. IR Activates Caspase-1/Interleukin-1 Converting Enzyme (ICE) in RAW264.7 Macrophages

IR increased the expression of caspase-1 at both the mRNA (Figure 3A) and protein (Figure 3B) levels in irradiated RAW264.7 macrophages compared with those in control cells. It was also confirmed that IR simultaneously increased caspase-1 protein expression and cleavage (Appendix A). In addition, the increase in caspase-1 activity by IR was also directly measured using a caspase-1 activity assay kit (Figure 3C). These results suggested that the IR-enhanced LPS-stimulated IL-1β production was at least partly regulated by the enzymatic activity of caspase-1. To confirm the role of the enzymatic activity of caspase-1 on the increase in IL-1β production by irradiation in LPS-treated RAW264.7 cells, we constructed a cell line in which caspase-1 was knocked down using the short hairpin RNA (shRNA). As shown in Figure 3D, Caspase-1 knockdown reduced the increase of IL-1β production in LPS-stimulated RAW264.7 macrophage cells compared with control shRNA-treated RAW264.7 macrophage cells, demonstrating a suppressed cytokine response to IR without caspase-1. The knockdown efficiency of caspase-1 using shRNA was verified using real-time PCR (Figure 3E) and a caspase-1 activity assay kit (Appendix A). IR also increased the expression of caspase-1 protein in J774A.1 macrophage cell (Appendix A). In addition, in J774A.1 cell, when caspase-1 was knocked down using shRNA, the increase in IL-1β production caused by IR was also suppressed (Appendix A). These results also indicate that the increase in IL-1β production by IR and the involvement of caspase-1 are not RAW264.7 cell-specific.

### 2.3. IR Activates the NLRC4 Inflammasome of RAW264.7 Macrophages

Formation of the inflammasome results in caspase-1 activation. To confirm the subtype of the inflammasome that regulates the enhanced production of IL-1β in irradiated RAW264.7 macrophages, the protein expression levels of the three inflammasome subtypes (NLRP3, NLRC4, and AIM2) were measured after IR. As shown in Figure 4A,B, irradiated RAW264.7 cells showed increased protein expression levels of NLRC4 inflammasome compared with control cells. By contrast, NLRP3 inflammasome protein was not expressed in either irradiated or control RAW264.7 cells. The protein expression level of AIM2 inflammasome did not differ between the two groups. In addition, the mRNA expression level of *Nlrc4* also significantly increased in the irradiated RAW264.7 macrophage cell line compared with that in the control group (Figure 4C). Activation of NLRC4 inflammasome in irradiated macrophages was further confirmed under confocal microscopy, with increased expression compared with that in the control (Figure 4D,E). However, there was no significant difference in the NLRC4 inflammasome expression level between LPS-treated RAW264.7 cells and control cells, which were both significantly lower than the levels in irradiated RAW264.7 cells. The expression level of NLRC4 inflammasome was the most significantly increased in IR and LPS-treated RAW264.7 cells. In addition, in the J774A.1 macrophage cell, the same confocal microscope image results were confirmed as in the RAW264.7 macrophage cell (Appendix A).

### 2.4. NLRC4 Plays an Important Role in IR-Induced IL-1β Production

Based on these results, we used small interfering RNA (siRNA) to silence the *Nlrc4* gene to confirm the role of the NLRC4 inflammasome in the increase in IL-1β production by irradiation in LPS-treated RAW264.7 cells. As shown in Figure 5A, irradiated cells treated with control siRNA showed a three-fold increase in IL-1β production in response to LPS stimulation compared with control cells. However, in NLRC4-knockdown macrophages, irradiation only resulted in a two-fold increase in the production of IL-1β following LPS stimulation compared with that in the control cells, demonstrating a suppressed cytokine response to IR without NLRC4. The same effects were found when measuring the protein levels of NLRC4 by western blotting (Figure 5B). These results provide direct evidence that activation of the NLRC4 inflammasome plays a vital role in the increased IL-1β production by irradiation.

### 2.5. Effects of Selective MAPK Inhibitors on IR-Induced IL-1β Production in Macrophages

Since the MAPK pathway plays an important role in the biological response of cells to radiation, we further investigated the MAPK that may be associated with increased IL-1β production in response to LPS stimulation in irradiated RAW264.7 macrophages. The total and phosphorylated forms of MAPK proteins were investigated 24 h after 5-Gy irradiation of macrophages. Irradiation increased the expression of phosphorylated p38 MAPK in RAW264.7 cells without affecting the expression of total p38 MAPK; however, there was no significant effect of irradiation on the phosphorylation of ERK or JNK proteins (Figure 6A). In J774A.1 macrophage cells, IR also increased the phosphorylation of the p38 MAPK protein (Appendix A).

The effect of selective inhibitors for each MAPK subtype on the increase in IL-1β production by IR + LPS was investigated. As shown in Figure 6A, the increase in IL-1β production by IR + LPS was significantly inhibited by the selective p38 MAPK inhibitor S8203580. However, neither the selective ERK (PD98059) nor JNK (SP600125) inhibitor affected the increase in IL-1β production caused by IR + LPS. As shown in Figure 6B,C, SB203580 inhibited the IR + LPS-enhanced IL-1β mRNA and intracellular protein expression levels in IR + LPS-treated RAW264.7 macrophages. However, SB203580 did not affect IL-1β production induced by LPS treatment only (Appendix A). These results suggested that the inhibitory effect of SB203580 is specific to the mechanism of IR but not that of LPS.

### 2.6. Effects of the Selective p38 MAPK Inhibitor on IR-Induced ROS Generation in Macrophages

IR increased the ROS level of RAW264.7 macrophage cells compared with that of the control (Figure 7A,B). However, treatment with the p38 MAPK inhibitor SB203580 significantly inhibited this IR-induced ROS production. In addition, the expression of heme oxygenase (HO)-1, another biomarker of oxidative stress, in RAW264.7 macrophage cells was increased following IR compared with that in the control group (Figure 7C). However, there was no significant difference in HO-1 protein expression between LPS-treated macrophages and control cells. IR + LPS treatment also significantly increased the expression of HO-1 protein, which was inhibited by SB203580.

## 3. Discussion

Radiation therapy is one of the major therapeutic modalities for most solid tumors. The anti-tumor effect of radiation therapy consists of direct tumor cell killing such as that due to DNA damage and ROS production, as well as modulation of the tumor microenvironment [33]. Macrophages represent an abundant immune cell population in the tumor microenvironment. Attempts have been made to determine whether IR can change the phenotypic characteristics of macrophages and if such changes can be controlled to improve therapeutic efficacy [34,35,36,37]. McKinney et al. [5] reported that IR can potentiate the production of NO induced by interferon (IFN)-γ and/or LPS in murine macrophages and indicated a possible role of tumor necrosis factor (TNF)-α in the process. Ibuki et al. [38] also reported that IR enhanced NO production via NF-kB activation. We previously reported that in addition to NO, IL-1β is selectively increased by irradiation in macrophages [12]. Considering the non-specific biological characteristics of IR or macrophages, this was an interesting novel finding. Furthermore, we found that IR-enhanced LPS-stimulated IL-1β increased the malignancy of CT26 colorectal cancer cells, thereby decreasing the CT26-implanted mice [13]. Adding to these previous research results, in the present study, we found that the molecular mechanism of the increased IL-1β production in response to LPS and IR exposure involves activation of the p38 MAPK-NLRC4 inflammasome-caspase-1 cascade pathway.

We did not classify macrophages as M1-type, M2-type, or M0-type macrophages in this study. This is because the RAW264.7 macrophage cell line is a widely used non-stimulated (naive, M0-type) macrophage cells such as PMA-treated THP-1 cells [39,40]. We conducted an experiment using syngenic RAW264.7 murine macrophage cells to reflect the complexity of the tumor environment and to maintain the intact immune environment of mice, which are widely used as experimental animals. Moreover, M1-type macrophage is not always associated with a good prognosis, and sometimes it is associated with a worse outcome than M2-type macrophage [41]. IL-1β, a typical cytokine secreted by M1-type macrophages, also does not necessarily act favorably in tumor treatment or as a biomarker for good prognosis [42]. In our previous study, increased production of irradiated macrophage-derived IL-1β also lowered the survival of CT26 colorectal cancer cell-bearing mice [13]. This is an intricate part to explain with the existing M1/M2 paradigm.

IL-1β is a potent inflammatory cytokine produced by macrophages, monocytes, and neutrophils, which exhibits multiple biological functions [43]. In cancer, IL-1β has pleiotropic effects on immune cells, cancer cell proliferation, and metastasis. In patients with advanced cancer, pro-inflammatory cytokines predominate, leading to the upregulation of IL-1 and increased production of downstream IL-6 [44,45]. Several studies reported that IL-1β also increases the stemness of colorectal cancer cells [46,47,48]. Radiation- or tumor-cell-induced IL-1β has been shown to promote the metastatic characteristics of tumor cells [49,50]. Recent clinical trials using a monoclonal antibody targeting IL-1β (canakinumab) also indicated a potential role of this cytokine in lung cancer [51]. These reports suggest that identifying the cause of aberrant IL-1β production and its regulation can be crucial in the prognosis of tumor treatment. Our findings suggest that macrophages can be an essential source of IL-1β under radiation therapy. According to our experimental results, the IR-induced elevation of IL-1β was much larger than that of NO production. These results suggest that IR has a more significant effect on IL-1β production than NO production of LPS-treated macrophages. In addition, to investigate the correlation between LPS-induced NO and IL-1β production induced by irradiation, the cells were pretreated with N-acetyl-L-cysteine (NAC), an NO scavenger, and IL-1β neutralizing antibodies. As a result, pretreatment with NAC did not inhibit the increase in IL-1β production by irradiation. Meanwhile, the neutralizing antibody of IL-1β also did not inhibit the increase in NO production by irradiation either (Appendix A). These results suggest that IR-induced increases in NO and IL-1β production in LPS-treated macrophages can be controlled independently of each other. IL-1β also can induce IL-6 production [52]. However, in our experiment, IR had no synergistic effect on the IL-6 production in LPS-treated macrophages.

The inflammasome is a multi-protein complex that serves as a molecular scaffold composed of adaptor molecules, a cytosolic pattern recognition receptor, and pro-caspase-1 [53], which plays an important role in the production of proinflammatory cytokines through the activation of caspase-1 in macrophages. The inflammasome consists of three main components: a sensor molecule, an adapter apoptosis-associated speck-like protein containing a caspase recruitment domain (CARD), and pro-caspase-1. NLRP3, absent in melanoma 2 (AIM2), and NLRC4 belong to the group of sensor proteins. Typically, caspase-1 is activated within the inflammasome that recognizes several damage-associated molecular patterns, which results in the cleavage of pro-IL-1β into mature IL-1β. IR can activate the inflammasome [19,20,54]. The main inflammasome type reported to be related to the production of IL-1β in macrophages is NLRP3; however, there are few reports on the production of IL-1β by the NLRC4 inflammasome. Under our experimental conditions, there was no increase in NLRP3 mRNA or protein expression induced by irradiation; among the inflammasomes, only the expression of NLRC4 was increased by irradiation. Kolb et al. [55] reported that obesity-induced NLRC4-mediated macrophage activation, thereby increasing IL-1β production. Another study indicated that NLRC4 and AIM2 are involved in ROS generation and DNA damage in cells induced by ultraviolet B irradiation [56]. Here, we provide the first report that the NLRC4 inflammasome is involved in IR-enhanced LPS-induced IL-1β production.

IL-1β and IL-18 are the only cytokines that are processed by caspase-1 after inflammasome-mediated activation. IL-1β exists in a pro-form in the cytoplasm and is cleaved as an active form by caspase-1 [57]; based on this function, caspase-1 is also known as interleukin-1 converting enzyme (ICE). Caspase-1 activation occurs after recruitment to the inflammasome. Caspase-1 also can induce apoptosis in response to various stimuli; for example, caspase-1 enhanced the apoptosis of DU-145 prostate cancer cells upon irradiation (0 to 9 Gy) [58]. IR can also induce caspase-1-dependent cell death [20,54]. Tabraue et al. [59] reported that IR induced the expression of caspase-1 in primary cultured bone marrow-derived macrophages. Our findings are in line with those of other previous studies, suggesting that IR can activate the enzymatic activity of caspase-1. In our experimental condition, caspase-1 activity and expression of macrophages were increased by irradiation (2 Gy). To our knowledge, this is the lowest radiation dose reported to date that resulted in increased caspase-1 activity. This suggests a high possibility of such activation in actual clinical situations. In some cases, IL-1β production occurred in a caspase-1-independent manner [60] and did not require TLR signaling [61].

Activation of MAPK plays a paramount role in various pathophysiological processes in macrophages [62,63,64]. ERK, p38 MAPK, and JNK are the three major subtypes of MAPK that play an important role in the production of pro-inflammatory cytokines in macrophages. These three MAPKs are also involved in various biological responses to irradiation [65,66]. Indeed, IR mediated JNK- and p38 MAPK-dependent innate immune cell activation [67]. Using inhibitors specific to each of these MAPK subtypes prior to IR, we found that the p38 MAPK inhibitor (SB203580) inhibited the IR-induced increase in IL-1β production both in RAW264.6 cell (Figure 6A) and J774A.1 cell (Appendix A). SB203580 also suppressed the IR-induced increase in the phosphorylation of p38 MAPK in RAW264.7 macrophages. The lack of change in the phosphorylation levels of ERK and JNK following irradiation suggests that p38 MAPK in macrophages can be an important target to increase IL-1β production by irradiation. In this regard, it has been reported that the p38 MAPK can regulate cellular responses to IR [67,68,69] and ultraviolet radiation [70]. Furthermore, Sun et al. [71] reported that CCN1 increased IL-1β production via p38 MAPK signaling in inflammatory disease. Furthermore, activation of the NLRC4 inflammasome could influence the phosphorylation of p38 [72]. None of the three MAPK inhibitors inhibited the increase in NO production by IR + LPS in macrophages (Appendix A). These results further suggest that the increased production of NO and IL-1β by IR+LPS can be regulated independently of each other. However, the effect of increased NO production by irradiation on cancer cells remains unknown. In general, NO is known to have a tumoricidal effect [73,74]. However, further studies are needed to determine the effect of irradiation on NO production in cancer cells under our experimental conditions. LPS is a widely used agonist to induce inflammation in macrophages. Induced nitric oxide synthase expression in macrophages is increased during LPS treatment, resulting in the activation of transcription factors such as NF-kB. The LPS used in our study can also phosphorylate MAPK, as well as ERK, JNK, and p38 MAPK, which typically occurs 30 min after treatment [75,76]. The p38 MAPK inhibitor SB203580 also has no inhibitory effect on LPS-induced p38 MAPK phosphorylation [76]. We found that IR only induces the phosphorylation of p38 MAPK but not ERK, and JNK. The p38 MAPK inhibitor SB203580 also inhibited the IR-induced p38 MAPK phosphorylation, NLRC4 expression, and ROS production in RAW264.7 macrophage cells. Together, these results suggest that p38 MAPK is an upstream target of IR-induced IL-1β production in LPS-stimulated RAW264.7 macrophage cells (Figure 8).

## 4. Materials and Methods

### 4.1. Cell Culture and Treatment

RAW264.7 murine macrophages were purchased from the American Type Culture Collection (ATCC, Manassas, VA, USA) and cultured at 37 °C in 5% CO_2_/95% air in Dulbecco’s modified Eagle’s medium (DMEM, Corning Inc., Corning, NY, USA) containing 10% heat-inactivated fetal bovine serum (Gibco; ThermoFisher Scientific, Waltham, MA, USA) and a penicillin (100 U/mL)/streptomycin (100 mg/mL) solution (Gibco; ThermoFisher Scientific, Waltham, MA, USA).

The cells (2 × 10^5^ cells/mL) were exposed to 5 Gy of IR with gamma rays from a Biobeam 8000 (137Cs source) (Gamma-Service Medical GmbH, Leipzig, Germany) at a dose rate of 2.5 Gy/min at room temperature. Following irradiation, the cells were incubated at 37 °C and then stimulated with LPS (0.1 μg/mL; Sigma-Aldrich, St. Louis, MO, USA) 24 h later.

### 4.2. Animals

C57BL/6NJ male mice (6 weeks old) were purchased from Central Lab. Animal Inc., (Seoul, Korea), and were allowed to acclimate to a specific pathogen-free (SPF) laboratory animal facility with free access to water and feed (Purina, Seoul, Korea). All animal protocols used in the current study were reviewed and approved by the Institutional Animal Care and Use Committee at Dongnam Institute of Radiological & Medical Sciences (DIRAMS; Busan, Korea) (Approval No. DIRAMS AEC-2018-013).

### 4.3. Preparation of Peritoneal Macrophages

Peritoneal exudates were obtained from C57BL/6 male mice (6 weeks old) by lavage 3 d after the intraperitoneal injection of 5 mL of 3% (*w*/*v*) Brewer’s thioglycollate medium (Sigma-Aldrich, Saint Louis, MO, USA) as reported previously [77]. After washing with RPMI-1640 medium containing 10% FBS, peritoneal macrophages (5 × 10^5^ cells/mL) were plated in 35-mm tissue culture dishes for 4 h at 37 °C in a 5% CO_2_ humidified atmosphere.

### 4.4. NO Assay

The culture supernatant was used for nitric dioxide (NO_2_^−^) determination using Griess reagent [0.1% N-(1-naphthyl)-ethylenediamine and 1% sulfanilamide in 5% phosphoric acid]. Equal volumes of culture supernatant and Griess reagent were mixed, and the absorbance was determined at 570 nm using a PARADIGM Detection Platform ELISA plate reader (Beckman Coulter, Fullerton, CA, USA). A normalized to the medium as a control.

### 4.5. Total RNA Isolation and Quantitative PCR

RNA was isolated from RAW264.7 cells using the RNeasy Mini Kit (Qiagen, Valencia, CA, USA), according to the manufacturer’s protocol. cDNA was synthesized with an iScript cDNA synthesis kit (170-8891, Bio-Rad, Hercules, CA, USA) so that 1 μg of RNA was added to a total reaction mixture of 20 μL on a Bio-Rad thermal cycler C1000 thermal cycler. PCR premixes were purchased from Bioneer (K-5051, Daejeon, Korea). The thermal cycling program consisted of a total of four stages: an initial denaturation step at 94 °C for 5 min, followed by 40 cycles of denaturation at 94 °C for 30 s, annealing at 55 °C for 30 s, and elongation at 72 °C for 30 s. The PCR mixture contained an SYBR Green fluorescent dye, enabling amplification detection and melt curve analysis. The melt curve measurement was monitored when it gradually increased by 1 °C per 15 s to 95 °C, starting with cultivation at 55 °C for 1 min. All quantitative PCR data were analyzed by the CFX96 Real-Time System (Biorad CFX384™ Real-Time System, Hercules, CA, USA) using Maxima SYBR Green qPCR master mix (K0251, Thermo Fisher Scientific, Waltham, MA, USA). Sequence information of all primers used in this study is indicated in Appendix A.

### 4.6. Western Blotting

The cells were collected using a scraper and treated with lysis buffer [150 mM NaCl, 20 mM Tris, pH 7.5, 25 mM EDTA, and 1% NP-40] with Halt™ Phosphatase Inhibitor Cocktail (ThermoFisher Scientific, Waltham, MA, USA). After mixing vigorously and maintaining on ice every 5 min three times, the cells were centrifuged at 13,000× *g* rpm at 4 °C for 15 min. After transferring the separated supernatant to a new tube, the protein concentration was determined by Bradford analysis. A total of 30 μg of protein was loaded onto a sodium dodecyl sulfate-polyacrylamide gel electrophoresis gel and then transferred to pre-chilled transfer buffer (0.25 M Tris, 1.92 M glycine, 20% methanol) using a 0.45-μm nitrocellulose membrane. The transferred membrane was blocked with 5% skim milk in Tris-buffered saline in Tween (TBST) for 1 h, and then the primary antibody was diluted at 1:1000 and reacted at 4 °C overnight. After the first antibody reaction, the membrane was washed three times with TBST, and the secondary antibody diluted at 1:5000 was reacted at room temperature for 1 h. The membrane was washed three times in the same manner and then developed using SuperSignal West Pico PLUS Chemiluminescent substrate (34580, ThermoFisher Scientific, Waltham, MA, USA). The catalog number, company, and dilution of all antibodies used in this study were listed in Appendix A.

### 4.7. Immunofluorescence Staining

The NLRC4 gene expression level of RAW264.7 cells was confirmed by confocal immunofluorescence staining. The sample was seeded in an eight-well chamber slide (354118, BD Biosciences, Franklin Lakes, NJ, USA) and fixed with 4% paraformaldehyde for 10 min. The slides were washed three times in 0.1% Triton-X 100 with DPBS, followed by incubation with primary antibodies diluted 1:1000 in 0.1% Triton X-100 and 1% bovine serum albumin for 1 h and washed in DPBS with 0.1% Triton X-100. The sample was then incubated with the secondary antibody (antibody information) diluted 1:500 in a dark room for 1 h. The cells were dropped in DAPI (4′,6-diamidino-2-phenylindole) on cover slides to immediately detect the fluorescence signal with a confocal microscope (Carl Zeiss LSM700 META, Oberkochen, Germany). The catalog number, company, and dilution of all antibodies used in this study were listed in Appendix A.

### 4.8. ELISA and Cytokine Arrays

RAW264.7 cell (~80% confluence) culture supernatants were collected after irradiation and LPS treatment, and cytokine expression levels were measured with ELISA kits (DY401, R&D Systems, Minneapolis, MN, USA) and Cytokine array (Proteome Profiler Mouse Cytokine Array kit, Panel (ARY006, R&D Systems, Minneapolis, MN, USA) according to the manufacturer protocols. The absorbance was measured at 450 nm on a microplate reader.

### 4.9. SiRNA Transfection

Transfections of scramble siRNA and mouse CARD12 (NLRC4) siRNA (Santa Cruz Biotechnology, Dallas, TX, USA) in RAW264.7 cells were performed using Lipofectamine RNAiMax transfection reagent (ThermoFisher Scientific, Waltham, MA, USA) according to the manufacturer’s protocol. The cells were seeded on a 60-mm dish at 2 × 10^5^ cells/mL one day before transfection. The medium was refreshed, gently mixed well with 100 nM of siRNA and 18 μL of Lipofectamine RNAiMax reagent, and then reacted for 15 min. The transfection mixture was slowly added dropwise onto the medium.

### 4.10. Cell Transfection and Lentiviral Infection

Lentivirus-based gene-specific small hairpin RNAs (shRNAs) were purchased from MISSION pre-designed mouse CASP1 shRNAs (Sigma-Aldrich, Saint Louis, MO, USA). Negative control shRNA (8453 for control plasmid pLKO.1, 12259 for envelop plasmid pMD2.G, and 12260 for packaging plasmid psPAX2)s were purchased from Addgene (Cambridge, MA, USA). Transfection was performed to 5 × 10^6^ HEK293T cells by using Lipofectamine^®^ 2000 Reagent (Invitrogen, Carlsbad, CA, USA) with 6.5 µg lentiviral vector, 5 µg psPAX2 and 2 µg pMD2.G. After 24 h, virus culture medium replaced with DMEM containing 10% FBS with 1% Antibiotic-Antimycotic. 48 h post-transfection, lentivirus-containing supernatant was collected and using filtered with a 0.45 µm syringe filter and concentrated using Lenti-X Concentrator (631232, Clontech; Takara-bio, San Jose, CA, USA). Target cell lines were transduced with 0.5 µg/mL of polybrene and lentiviral inducible RNAi at MOI = 1 and selected in 2 µg/mL Puromycin Dihydrochloride (A1113802, Gibco; ThermoFisher Scientific, Waltham, MA, USA). The culture medium was changed 24 h post-infection and target cells were harvested 48 h after puromycin selection.

### 4.11. Measurement of Direct Caspase-1 Activity Using Culture Supernatant

The caspase-1 activity was performed using Caspase-Glo^®^ 1 Inflammasome Assay kit (G9951, Promega, Madison, WI, USA) according to the manufacturer’s protocol. In brief, the Z-WEHD-aminoluciferin caspase-1 substrate was mixed with Caspase-Glo^®^ 1 Buffer and equilibrated to room temperature before use. Then, 100 μL of culture supernatant was reacted with an equal volume of substrate solution and was added to each well of the white-walled 96-well plate. The plates were gently mixed using a plate shaker at 300–500 rpm for 30 s and then incubated at room temperature for at least 1 h to allow the luminescent signal to stabilize. Luciferase activity was measured with the Promega GloMax Discover microplate reader (Promega, Madison, WI, USA).

### 4.12. ROS Detection

RAW264.7 cells (2.5 × 10^5^ cells/mL) were incubated 24 h after irradiation and cultured for an additional 24 h after LPS treatment. The remaining culture medium was removed, and the cells were washed with PBS. Intracellular ROS levels were measured using the fluorescent dye 2′,7′-dichlorodihydrofluorescein diacetate (DCFDA; Sigma-Aldrich, Saint Louis, MO, USA), followed by treatment with 1.25 mM of DCFDA in a dark room for 30 min.

### 4.13. Statistical Analysis

Data are represented as the mean ± SEM of at least three independent experiments, performed in triplicate. Student’s *t*-test was carried out to analyze the statistical significance between the groups using SPSS version 18.0 (SPSS Inc., Chicago, IL, USA). A *p* value < 0.05 was considered statistically significant.

## 5. Conclusions

In conclusion, we revealed that IR increases the NLRC4 inflammasome and caspase-1 activity, which governs the conversion of premature IL-1β to soluble IL-1β, in murine macrophages. We further confirmed that p38 MAPK is the primary regulator of IR-induced activation of the NLRC4 inflammasome complex. Thus, the NLRC4-caspase-1-p38 MAPK axis could be a potential molecular target for improving the outcome of radiotherapy.

## Figures and Tables

**Figure 1 ijms-23-13757-f001:**
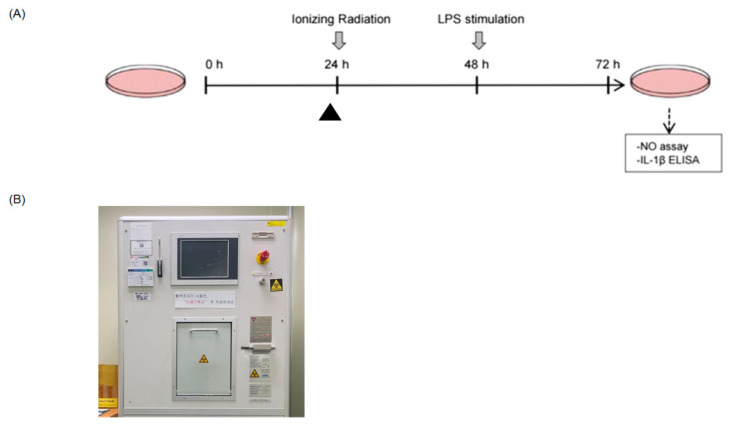
Schematic diagram of the experimental design. (**A**) RAW264.7 macrophages (2 × 10^5^ cells/mL) were exposed to 5 Gy of IR and then stimulated with LPS (0.1 μg/mL) 24 h later. After incubation for another 24 h, the supernatant was recovered to evaluate the IL-1β and nitrite (NO_2_^−^) levels secreted from RAW264.7 cells. The arrowhead (▲) indicates pretreatment with a signal transduction inhibitor 30 min before irradiation. (**B**) The medical blood irradiator used for cell irradiation.

**Figure 2 ijms-23-13757-f002:**
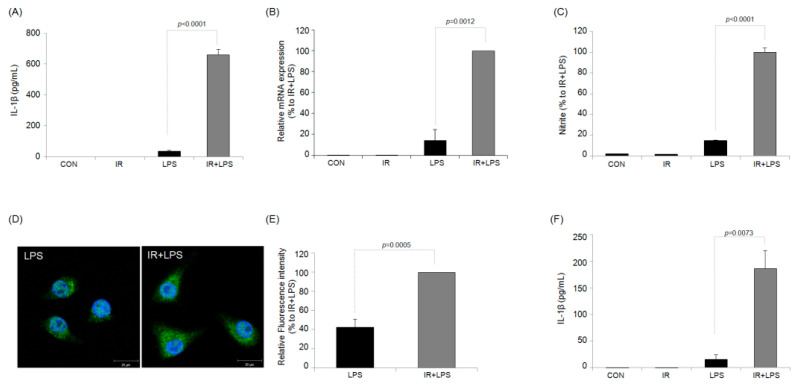
Ionizing radiation (IR) potentiates the IL-1β production of RAW264.7 macrophages and primary cultured peritoneal macrophages in response to lipopolysaccharide (LPS). RAW264.7 (2 × 10^5^ cells/mL) cells were irradiated (5 Gy) using a blood gamma irradiator. After 24 h, the cells were treated with LPS (0.1 μg/mL). After another 24 h, the supernatant was used for IL-1β (**A**) or nitrite (NO_2_^−^) (**C**) determination. (**B**) Measurement of the mRNA level of IL-1β in the different treatment groups. (**D**,**E**) Representative confocal microscopy image for intracellular IL-1β detection of LPS only- and IR + LPS-treated macrophages and fluorescence analysis data. (**F**) Primary cultured peritoneal macrophage (5 × 10^5^ cells/mL) cells were irradiated (5 Gy) using a blood gamma irradiator. After 24 h, the cells were treated with LPS (0.1 μg/mL). After another 24 h, the supernatant was used for IL-1β.

**Figure 3 ijms-23-13757-f003:**
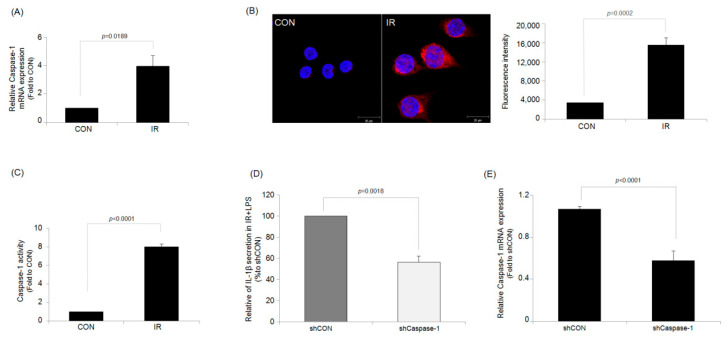
Ionizing irradiation (IR) increases the caspase-1 activity of RAW264.7 macrophage cells. (**A**) mRNA and (**B**) protein expression of caspase-1 in RAW264.7 macrophages. (**C**) Changes in caspase-1 activity were measured. (**D**) Changes in IL-1β production in RAW264.7 macrophage cells transfected with either caspase-1 shRNA or scramble shRNA. (**E**) Real-time PCR data of caspase-1 mRNA. Results are expressed as mean ± S.E.M.

**Figure 4 ijms-23-13757-f004:**
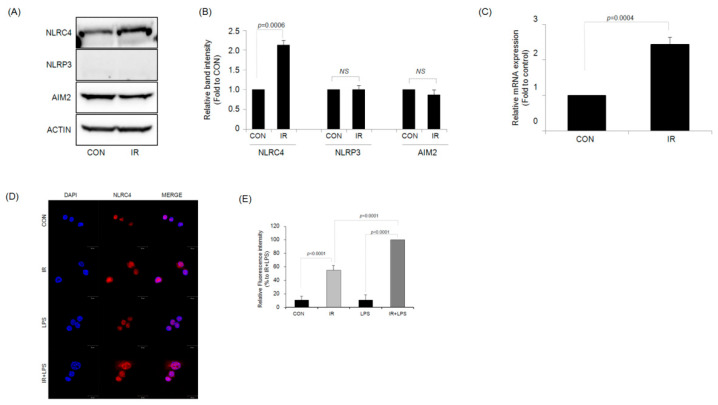
Ionizing radiation (IR) activates the NLRC4 inflammasome in the RAW264.7 macrophage cell line. (**A**) Representative western blotting images and (**B**) quantitative analysis of inflammasome protein levels in control and irradiated RAW264.7 macrophages. (**C**) Analysis of *Nlrc4* mRNA levels by quantitative PCR in control and irradiated RAW264.7 macrophages. (**D**,**E**) Fluorescence images of RAW264.7 cells after staining with NLRC4; red (PE) indicates NLRC4, and blue (DAPI) indicates the nuclei. Scale bar = 20 μm.

**Figure 5 ijms-23-13757-f005:**
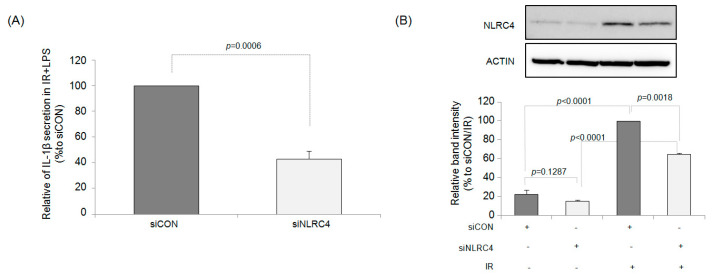
siNLRC4 transfection suppressed the increase in IL-1β production by irradiation in LPS-stimulated macrophages. (**A**) Changes in IL-1β production in RAW264.7 macrophage cells transfected with either NLRC4 siRNA or scramble siRNA. (**B**) Representative western blotting image of NLRC4 protein expression with relative band intensity quantification data (% to IR treated siCON). Results are expressed as mean ± S.E.M.

**Figure 6 ijms-23-13757-f006:**
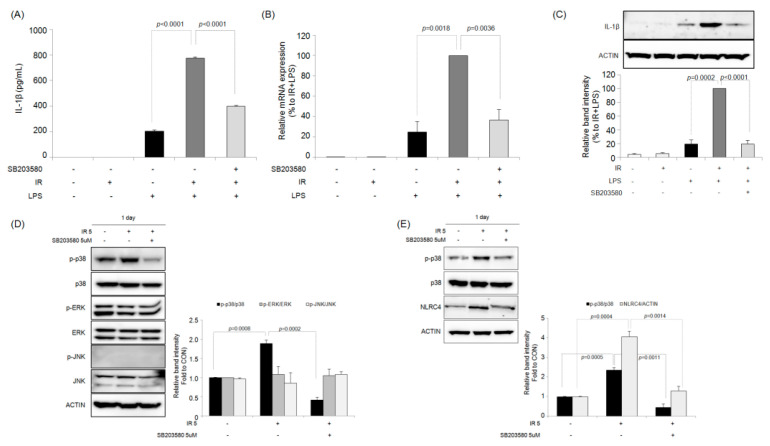
The p38 MAPK inhibitor SB203580 suppresses the ionizing radiation (IR)-enhanced LPS-stimulated IL-1β production in RAW264.7 macrophages. (**A**) RAW264.7 macrophages were pretreated with ERK, JNK, and p38 MAPK inhibitors as indicated for 30 min, followed by IR (5 Gy). The cells were treated with LPS 24 h later, and IL-1β was quantified after a further 24 h of incubation. Results are expressed as mean value ± S.E.M. The expression level of IL-1β was analyzed using quantitative real-time PCR (**B**) and western blotting (**C**). (**D**) SB203580 inhibits the phosphorylation of p38 MAPK induced by IR in RAW264.7 macrophages. (**E**) SB203580 also inhibits the expression of NLRC4 induced by IR in RAW264.7 macrophage cells.

**Figure 7 ijms-23-13757-f007:**
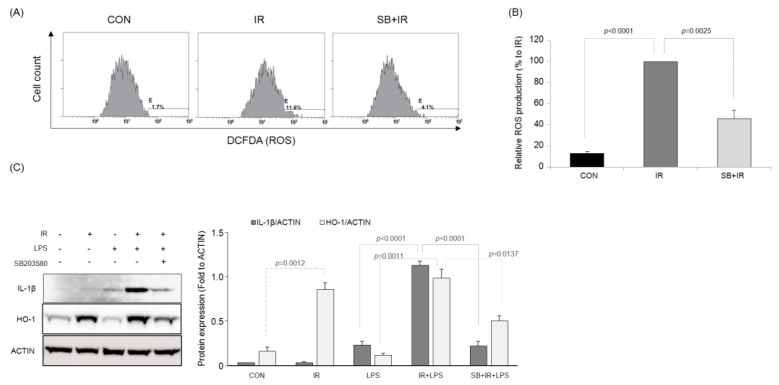
The p38 MAPK inhibitor SB203580 inhibits the ionizing radiation (IR)-mediated reactive oxygen species (ROS) production in RAW264.7 macrophages. (**A**,**B**) Production of ROS. (**C**) HO-1 protein expression. Bar graph results are expressed as mean value ± S.E.M.

**Figure 8 ijms-23-13757-f008:**
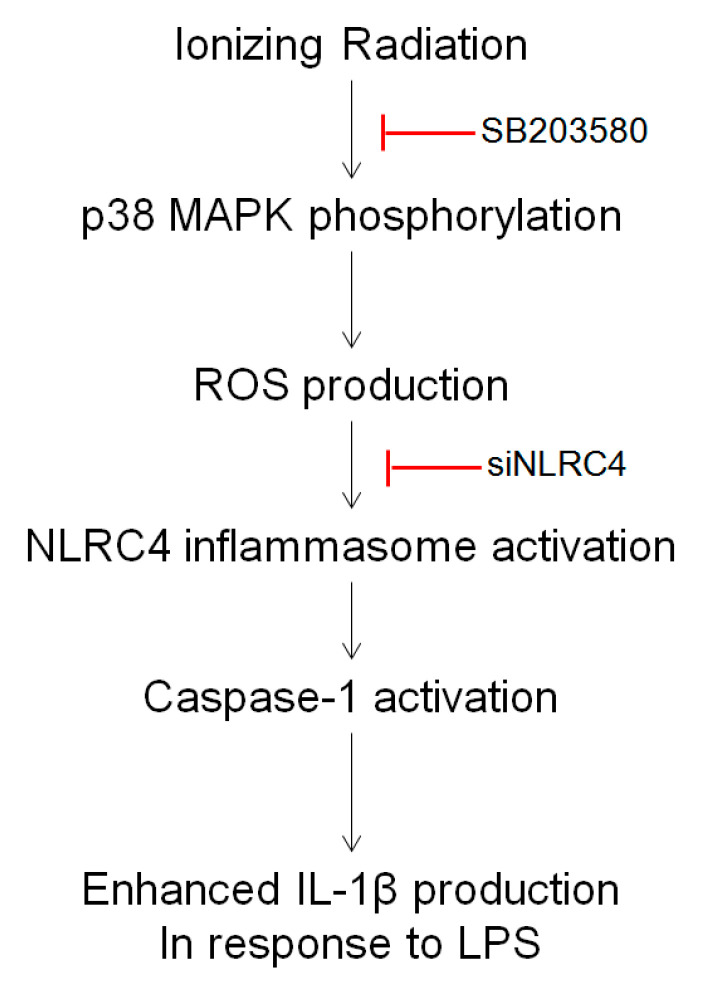
Proposed model for the molecular mechanism of IR-enhanced LPS-stimulated IL-1β production in RAW264.7 macrophage cells.

## Data Availability

Not applicable.

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
