# Peer review of "Involvement of the p38 MAPK-NLRC4-Caspase-1 Pathway in Ionizing Radiation-Enhanced Macrophage IL-1β Production"

_ijms, 2022, doi:10.3390/ijms232213757_

Round 1
Reviewer 1 Report
Revision of the manuscript “ijms-1940216” by Baik et al. The manuscript has an interesting proposal; however, there are some issues to be addressed before acceptance, as follows:
Major points:
1) There are critical limitations regarding the model used. First, as well described in the literature, M1 and M2 phenotypes have opposite roles in the cancer context; however, the authors did not induce any differentiation to evaluate it. I recommend checking Martinez et al. (doi: 10.12703/P6-13. eCollection 2014), Murray et al. (doi: 10.1016/j.immuni.2014.06.008) and Lv et al (doi: 10.3892/mmr.2017.7719), for instance, to conduct these differentiations and perform the experiments comparing M1 and M2 phenotypes. Second, the authors have only used RAW 264.7 cells, which are mouse macrophages, as a model. I strongly recommend including human macrophages; ideally, primary macrophages (you can find some options commercialized by Lonza and StemCell Tech) or at least monocytic cell lines, such as THP-1, differentiated into macrophages.
2) Figure 3 shows increased mRNA levels of Caspase-1 in IR cells. As it has been done in their previous paper (Baik et al. 2020), I recommend showing a full blot of Caspase-1 to evidence its cleavage forms.
3) On page 4, lines 144-146, the authors stated that “These results suggested that the IR-enhanced LPS-stimulated IL-1β production was at least partly regulated by the enzymatic activity of caspase-1.” It is necessary to knockdown Caspase-1 to confirm this statement, which I strongly recommend.
Minor points:
1) The Immunoblotting pictures are too small, especially on Figure 6. It makes the results hard to understand.
2) Please include the data not shown (page 213, line 210) on the Supplementary file.
3) The sequence and the specific melting temperature of the primers used in this manuscript need to be described in Methods. I recommend showing this information in a Supplementary table.
4) The catalog number, company, and dilution of all antibodies used in this manuscript need to be included in Methods.
Author Response
First, I would like to thank the reviewers for their opinions. Through this opportunity, we could look back on the results of our research and get a clue as to the direction we need to move forward. While contemplating the reviewer's comments and finding answers, we felt that our manuscript had developed scientifically.
Here we reply to the reviewer's suggestions point-to-point in the attached response sheet.
The authors faithfully carried out the reviewer's suggestions during this review period. Although not all of the reviewer's suggestions were met, it is judged that our manuscript has developed more scientifically through this review. Thank you again.

Reviewer 2 Report
The authors explored the ionizing radiation (IR) increased the expression of caspase-1 and IL-1β in RAW264.7 macrophages. Furthermore, irradiated raw264.7 cells increased expression of NLRC4 inflammasome, which is the upstream of caspase-1 and IL-1β. In mechanism, the authors use siRNA and inhibitors to conclude that p38 MAPK is the primary regulator of IR-induced activation of the NLRC4 inflammasome complex. Therefore, the NLRC4-caspase-1-p38 MAPK axis could be a potential molecular target for improving the outcome of radiotherapy.
However, there are still some issues need to be addressed before publication.
1. In Figure 2D, the authors can quantify the fluorescence intensity;
2. In Figure 5B, the authors can quantify the WB result;
3. In Figure 6C, D and E, in Figure 7C also need to be quantified the WB results;
4. In Figure 8, the authors can add the siNLEC4 and P38 MAPK inhibitor on the panel to make the proposed model more completed;
5. In Figure 2D, 3B, there needs a bar scale on the panels.
6. Please have a list for the primers the authors used in PCR experiment.
Author Response
Thank you for the reviewer's heartwarming encouragement and comments. Authors reply to the reviewer's suggestions in a point-to-point manner in the attached response sheet.
The authors faithfully carried out reviewer 2's suggestions during this review period.
In the process of responding to the reviewers' suggestions, it has been a great help in strengthening the logic of our manuscript and helping readers understand it.
Thank you again.

Round 2
Reviewer 1 Report
The second round of revision of the manuscript “ijms-1940216” by Baik et al. The minor points have been addressed; however, the major points are still in the current version:
Major points:
1) In my previous letter, I described the relevance of understanding the M1 and M2 phenotypes in the proposed authors’ context, and they stated in the cover letter that “it is judged that it is crucial to find out what kind of difference in the response of macrophages differentiated into M1 and M2 cells to irradiation”. However, I could not observe any improvement on this topic.
Regarding the models, I am glad that the authors understood the importance of using human macrophages in their studies and have purchased the suggested cells. Still, demonstrating that the proposed axis is not RAW-cells-specific is mandatory. Thus, I strongly recommend the authors conduct the experiments on the purchased primary macrophages or, at least, on differentiated THP-1 cells.
2) On the second topic of the cover letter, the authors addressed that “(…) a caspase-1 knockdown experiment using shRNA technology confirmed that the increase in IL-1β production by IL+LPS was suppressed without caspase-1 (Figure 3D and 3E).” Including the knockdown model was very relevant. However, it was not clear how the caspase 1 activity was measured. I did not understand why the full blot could not be performed if the authors had done it in a previous paper (Baik et al. 2020). Still, showing qPCR results for Caspase-1 is not enough since the mature/cleaved isoforms cannot be precisely detected. An alternative is using an assay kit, such as:
https://www.abcam.com/caspase-1-assay-kit-colorimetric-ab273268.html
https://www.promega.com.br/products/cell-health-assays/inflammation-assay/caspase_glo-1-inflammasome-assay?gclid=CjwKCAjw7p6aBhBiEiwA83fGuklZjXtw-TRffL62LDmJEp4rYIh8LNhsy81AIra-Pp0MwCFYkYuRLBoCaxMQAvD_BwE&catNum=G9951
https://www.rndsystems.com/products/caspase-1-ice-colorimetric-assay-kit_k111-100#product-details
I recommend the same for the third topic of the cover letter – regarding Figures 3D and 3E.
Minor points:
1) The blots on figs 6D and 6E are still too small.
Author Response
We would like to appreciate our sincere gratitude to Reviewer 1 for their time and effort for the development of this study.
By following Reviewer 1's advice and suggestions, we were able to add more scientific rationale to our findings.

Reviewer 2 Report
The issues i concerned were addressed.
Thank you.
Author Response
We would like to appreciate our sincere gratitude to Reviewer 2 for their time and effort for the development of this study.
Round 3
Reviewer 1 Report
The third round of revision of the manuscript “ijms-1940216” by Baik et al.
Minor points:
1) The statement “(…) the authors did not classify macrophages as M1-type, M2-type, or M0-type macrophages in this study” mentioned in the last cover letter needs to be included in the Discussion section with the literature references that the authors have chosen to support their decision.
2) The legend of the Supplementary Figure 1 needs to be reviewed because “Even in (…)” does not sound scientific.
3) The figure identified as “Figure 1E. Ionizing radiation induces increased IL-1β production for LPS in primary cultured peritoneal macrophage cells” on the cover letter needs to be included in the manuscript.
4) Please check if all blots were included in the Original images file. For instance, the blot shown in Supplementary Figure 4A is not there.
Author Response
We would like to appreciate our sincere gratitude to Reviewer 1 for their time and effort for the development of this study again.
We were able to add more scientific rationale to our findings.
